# Interactions between Ambient Air Particles and Greenness on Cause-specific Mortality in Seven Korean Metropolitan Cities, 2008–2016

**DOI:** 10.3390/ijerph16101866

**Published:** 2019-05-27

**Authors:** Sera Kim, Honghyok Kim, Jong-Tae Lee

**Affiliations:** 1BK21PLUS Program in ‘Embodiment: Health-Society Interaction’, Department of Public Health Science, Graduate School, Korea University, Seoul 02841, Korea; ssera0905@gmail.com (S.K.); honghyok@korea.ac.kr (H.K.); 2School of Health Policy and Management, College of Health Science, Korea University, Seoul 02841, Korea

**Keywords:** air pollution, particulate matter, greenness, Normalized Difference Vegetation Index, mortality, urban environment

## Abstract

This study aims to investigate the association of particulate matter with an aerodynamic diameter smaller than 10 μm (PM_10_) and greenness with cause-specific mortality and their interactions in seven Korean metropolitan cities. We obtained the annual standardized cause-specific mortality rates, annual mean concentration of PM_10_, and annual Normalized Difference Vegetation Index (NDVI) for 73 districts for the period 2008–2016. We used negative binomial regression with city-specific random effects to estimate the association of PM_10_ and greenness with mortality. The models were adjusted for potential confounders and spatial autocorrelation. We also conducted stratified analyses to investigate whether the association between PM_10_ and mortality differs by the level of greenness. Our findings suggest an increased risk of all causes examined, except respiratory disease mortality, with high levels of PM_10_ and decreased risk of cardiovascular-related mortality with a high level of greenness. In the stratified analyses, we found interactions between PM_10_ and greenness, but these interactions in the opposite direction depend on the cause of death. The effects of PM_10_ on cardiovascular-related mortality were attenuated in greener areas, whereas the effects of PM_10_ on non-accidental mortality were attenuated in less green areas. Further studies are needed to explore the underlying mechanisms.

## 1. Introduction

Residential environments have significant impacts on health. Air pollution has been associated with an increased risk of cardiopulmonary health, including cardiovascular disease, respiratory disease, and lung cancer [1,2,3,4]. Conversely, green space has been recently suggested to benefit health, especially cardiovascular disease, mental health, and levels of overweight and obesity [5,6,7].

However, the health effects of these environmental factors may not appear to be independent of one another. Green space has been suggested to promote health by encouraging physical activity [8,9], facilitating social contacts [10], and decreasing psychological stress [11,12], which may affect the response to air pollution. Therefore, when people are simultaneously exposed to air pollution and greenness, the detrimental effects of air pollution may, to some degree, be mitigated by the beneficial effects of greenness.

Few studies have investigated such interactions between air pollution and greenness, and those results are inconsistent. Kioumourtzoglou et al. [13] found larger effects of particulate matter on all-cause mortality in greener areas. de Keijzer et al. [14] also found stronger effects of particulate matter on all-cause mortality in urban areas with higher greenness, while stronger effects were observed in rural areas with lower greenness. MacNaughton et al. [15] found a larger effect of particulate matter on school absenteeism in greener areas. These limited results require further research that evaluates the risk of air pollution and greenness simultaneously with various health outcomes in a diverse population.

South Korea, where 91.8% of the population lives in urban areas [16], is one of the highly urbanized countries with poor air quality and less green areas. However, no study has so far examined the effects of combined exposure to air pollution and greenness. Therefore, this study aimed to investigate the association of ambient air particles and greenness with cause-specific mortality and interactions between two environmental factors in seven Korean metropolitan cities.

## 2. Materials and Methods

### 2.1. Data

This study investigated the effects of particulate matter with an aerodynamic diameter smaller than 10 μm (PM_10_) and greenness on cause-specific mortality in seven metropolitan cities (Seoul, Busan, Daegu, Incheon, Gwangju, Daejeon, and Ulsan) in Korea during the period 2008–2016. We used a local district (gu/gun) as the unit of analysis. Gu or gun, which will be referred to hereafter as the district, is an administrative unit that is similar to borough in European and American cities, and the smallest geographic unit for which national statistical data is collected in Korea. There is a total of 74 districts in seven metropolitan cities, and exclusion was made for the district with missing information on air pollution (Ongjin-gun in Incheon City). Information on the total population, area, and urbanization rate for each district was obtained from the Korean Statistical Information Service.

We obtained the data on all registered deaths from the Korean National Statistical Office for the period 2008–2016. A priori selected causes were non-accidental (ICD10; A00-R99), cardiovascular disease (I00-99), ischemic heart disease (I20-25), respiratory disease (J00-99), chronic lower respiratory disease (J40-47), and lung cancer (C33-34). Lung cancer was included as a negative control outcome for greenness because no clear plausible mechanism has been demonstrated [17,18]. The annual age and sex-standardized mortality rates were calculated for each district by direct standardized methods using the 2005 Census as the standard population.

We obtained hourly measurements of PM_10_ from the National Institute of Environmental Research for the study period. There were one or two monitoring stations measuring air pollutants including PM_10_ in each district. If a district had more than one monitoring stations, we calculated the representative value by averaging hourly measurements of the stations. We then calculated the annual mean concentration of PM_10_ from daily 24-h mean concentrations for each district. If more than 25% of the hourly measurements of a station were missing, we substituted the daily mean concentration with an annual average concentration from valid daily estimates of each monitoring station in that year.

The greenness was indicated by the Normalized Difference Vegetation Index (NDVI), which is a widely used indicator to quantify green vegetation. It ranges from –1 to 1 and higher NDVI values indicate greener areas [19]. The 16-day composite NDVI dataset at 250 m resolution was obtained from the Moderate Resolution Imaging Spectroradiometer (MODIS) from National Aeronautics and Space Administration (NASA)’s Terra satellite from 2008 through 2016 (MOD13Q1 product) [20]. We averaged non-negative NDVI values of all pixels within the administrative boundary for each district and then used the median value of NDVI for the summer period (May–October), which indicates the level of greenness in peak bloom.

We included neighborhood socioeconomic status, smoking rates, and healthcare infrastructure status as potential confounders. Education level was used to indicate neighborhood socioeconomic status, because education is one of the indicators that reflect the complex dimensions of socioeconomic status in terms of not only social, psychological, and economic resources, but also health knowledge acquisition [21,22]. We measured the percentage of adults with low education using the Population and Housing Census. Low education was defined as: (1) less than high school education for those aged under 45, and (2) less than middle school education for those aged 45 or over [23]. By using a different standard of low education according to age groups, we intended to avoid the problem that older population, whose educational attainment are relatively lower than younger population, may be overrepresented as less educated. The smoking rates were obtained from the 2008–2016 Korean Community Health Survey, which is a cross-sectional representative national survey. The health care resource index was used to indicate neighborhood healthcare infrastructure status, using data from the 2008–2016 National Health Insurance Service. This index consisted of the numbers of medical personnel, hospitals, and hospital beds (per 100,000 population), which are widely used indicators to represent and compare health care resources across countries [24]. We converted the value on each indicator to a z-score and used the sum of the z-scores as the health care resource index. Therefore, the index has a mean of 0 and a higher value of the index indicates a higher utilization of health care resources.

### 2.2. Statistical Analysis

We analyzed time-series data at an annual level [25] to assess the association of PM_10_ and greenness with cause-specific mortality. We used negative binomial regression to account for the over-dispersion of count data with city-specific random effects. We used single- and two-exposure models with adjustment for neighborhood socioeconomic status, smoking rate, and healthcare infrastructure status. In the single-exposure model, we assessed the effects of PM_10_ and greenness in separate models. In the two-exposure model, we assessed the effects of each exposure in the mutually adjusted model. Spatial autocorrelation of residuals was tested using Moran’s I statistics. As we found significant residual spatial autocorrelation in all models, we additionally adjusted for the coordinate of each district in the model. We used the coordinates of each district’s office station.

To investigate whether the association between PM_10_ and mortality varied by the level of greenness, we included an interaction term between PM_10_ and greenness in the model and stratified the analyses. Greenness was categorized into tertiles (<33%, 33% to <66%, and ≥66%) to obtain sufficient data within each category and practical interpretation (low/medium/high) of values.

Additionally, we conducted the sensitivity analyses by: (1) adjusting for gaseous pollutants (SO_2_ and NO_2_), and (2) using different indicators of neighborhood socioeconomic status, including median annual household income, local tax per capita, and percentage of those who receive social benefits, instead of the percentage of adults with low education. We obtained the data for these indicators from the 2008–2016 Korean Community Health Survey and the Korean Statistical Information Service. We also conducted city-specific analyses to explore spatial variations in the association with PM_10_ and greenness with mortality. The results for the association of PM_10_ and greenness with mortality were presented for 10 μg/m^3^ and interquartile range (IQR) increase, respectively. All analyses were conducted using SAS software version 9.4 (SAS Institute, Cary, NC, USA) and ArcGIS 10.3.1 (ESRI, Redlands, CA, USA).

## 3. Results

### 3.1. Descriptive Statistics

This study included 73 districts with an average population per district of 317,869 and the size of 71.63 km^2^ in 2012. These districts are highly urbanized areas with an average urbanization rate, defined as the percentage of population living in urban areas, of 97.72%. Table 1 presents the descriptive statistics of the study variables. The average of annual mean concentration of PM_10_ was 47.65 μg/m^3^ and the annual mean NDVI was 0.48. The average percentage of adults with low education, smoking rate, and health care resource index were 14.13%, 23.97%, and 0.00, respectively. Districts with high levels of PM_10_ tended to have lower NDVI values and lower education levels (Appendix A). The spatial distributions and annual trends of exposure variables and standardized mortality rates are presented in Appendix A.

### 3.2. Association of PM_10_ and Greenness with Cause-Specific Mortality

Table 2 shows the association of PM_10_ and greenness with cause-specific mortality, controlling for potential confounders and spatial autocorrelation. PM_10_ was significantly associated with increased risks of all causes examined, except respiratory disease mortality. In the models with an adjustment for greenness, a 10 μg/m^3^ increase in annual average PM_10_ was associated with 4.49% (95% confidence interval (CI): 3.41%, 5.57%) increase in non-accidental, 9.70% (95% CI: 7.64%, 11.81%) in cardiovascular disease, 7.50% (95% CI: 4.19%, 10.90%) in ischemic heart disease, 16.03% (95% CI: 11.42%, 20.85%) in chronic lower respiratory disease, and 2.98% (95% CI: 0.92%, 5.08%) in lung cancer mortality. Conversely, PM_10_ was associated with 3.12% (95% CI: −5.36%, –0.83%) decrease in respiratory disease mortality.

Greenness was significantly associated with decreased risks of cardiovascular disease and ischemic heart disease mortality (Table 2). An IQR increase in NDVI was associated with 2.89% (95% CI: –5.18%, −0.53%) decrease in cardiovascular disease and 3.64% (95% CI: −7.08%, −0.06%) decrease in ischemic heart disease mortality. In the models with an adjustment for PM_10_, the associations with mortality were slightly attenuated. We observed the corresponding estimates of −2.56% (95% CI: −4.68%, −0.39%) and −3.45 % (95% CI: −6.84%, 0.07%) for cardiovascular and ischemic heart disease mortality with adjustment for PM_10_, respectively.

### 3.3. Interactions Between PM_10_ and Greenness

The associations between PM_10_ and mortality varied by the level of greenness. However, greenness modified the associations differentially depending on the cause of death (Table 3). The association between PM_10_ and non-accidental mortality was stronger in districts with a higher level of greenness. The observed estimates were 5.83% (95% CI: 3.95%, 7.74%), 3.57% (95% CI: 1.81%, 5.37%), and 3.45% (95% CI: 1.42%, 5.51%) for high, medium, and low levels of greenness, respectively. On the other hand, the association between PM_10_ and ischemic heart disease mortality was stronger in districts with a lower level of greenness. The observed estimates were 1.89% (95% CI: −4.51%, 8.72%), 6.73% (95% CI: 1.58%, 12.13%), and 7.86% (95% CI: 1.52%, 14.60%) for high, medium, and low levels of greenness, respectively. Similar trends were found in cardiovascular mortality across the level of greenness. However, the interaction terms in cardiovascular and ischemic heart disease mortality were not statistically significant (*p* = 0.67 and 0.10, respectively).

The associations between PM_10_ and cause-specific mortality were robust when either SO_2_ or NO_2_ was additionally adjusted for (Appendix A). Furthermore, using different indicators of neighborhood socioeconomic status did not affect the association of PM_10_ and greenness with cause-specific mortality and their interactions (Appendix A). In the city-specific analyses, we found different patterns of PM_10_ or greenness with mortality across cities (Appendix A). Daejeon was found to have a negative estimate in the association with PM_10_ and cardiovascular and respiratory-related mortality. Additionally, Daegu and Ulsan were found to have positive, but non-significant estimates in the association with greenness and ischemic heart disease mortality.

## 4. Discussion

We investigated the association of PM_10_ and greenness with cause-specific mortality and the interactions between two environmental factors in seven Korean metropolitan cities. We found an increased risk of all causes examined, except respiratory disease mortality, with high levels of PM_10_ and decreased risk of cardiovascular-related mortality with a high level of greenness. We also found interactions between PM_10_ and greenness on mortality, although their relationship differed depending on the cause of death. The effect of PM_10_ on non-accidental mortality was stronger in districts with a higher level of greenness. On the other hand, the effect of PM_10_ on cardiovascular mortality was stronger in districts with a lower level of greenness, although interaction terms were not significant.

The overall literature has suggested the increased risks of all-cause and cardiovascular mortality with increment of PM [3,4]. While the underlying biological mechanisms are unclear, exposure to ambient particulate matter has been related to the occurrence of pulmonary and systemic oxidative stress and inflammation, perturbation of autonomic nervous system balance, and translocation of particulate matter directly into the systemic circulation [1,4]. A meta-analysis by Hoek et al. [3] reported the excess risk of 3.5% (95% CI: 0.4%, 6.6%) per 10 μg/m^3^ increase in PM_10_ for all-cause mortality. A recent study, a large cohort study with long-term follow-up, also reported that long-term exposure to PM_10_ was associated with the increased risk of all-cause mortality (Hazard Ratio (HR) (95% CI) = 1.03 (0.94, 1.14) per 10 μg/m^3^ increment of PM_10_) and cardiovascular mortality (HR (95% CI) = 1.23 (1.02, 1.49)) [26]. Our findings regarding the adverse effects of PM_10_ on chronic lower respiratory disease and lung cancer mortality are also consistent with the results of previous studies [2,27]. However, contrary to our predictions, we found a negative association between PM_10_ and respiratory disease mortality. Although several studies have reported the lack of association between long-term exposure to PM and respiratory mortality [26,28,29], these findings do not fully explain our negative findings. One possible explanation is an insufficient adjustment for potential confounders in the association between PM_10_ and respiratory mortality. The majority of the observed respiratory mortality was from pneumonia (59.3%), but we could not consider risk factors other than air pollution, such as living environment causing infection to bacteria, or underlying diseases, which might be negatively correlated with air pollution. Given that PM_10_ was significantly associated with increased chronic lower respiratory disease mortality, which is a subcategory of respiratory mortality, further research with more disease-specific respiratory mortality and considering other potential confounders is needed to address the uncertainty over this finding.

Other findings of the protective effects of greenness are also in line with previous research. In a prospective cohort study with a large and nationally representative sample, an IQR increase in NDVI was associated with reduced non-accidental (HR (95% CI) = 0.92 (0.91, 0.92)), cardiovascular disease (HR (95% CI) = 0.91 (0.89, 0.93), ischemic heart disease (HR (95% CI) = 0.90 (0.88, 0.93), and respiratory disease (HR (95% CI) = 0.90 (0.87, 0.93)) mortality [30]. Several meta-analyses have reported the protective effects of green space on all-cause and cardiovascular mortality [5,6,7]. Despite differences in study design and study population, the results of previous studies were generally similar to ours. We also selected lung cancer as the negative control outcome for greenness to detect confounding and other biases [31], and the lack of association with exposure to green space lends confidence to the validity of our findings. We also additionally adjusted for PM_10_ to control for its indirect effects on health impacts. After adjusting for PM_10_, the associations between greenness and mortality were slightly attenuated, which indicates that PM_10_ is responsible for some of the effects of greenness on mortality. Greenness is suggested to reduce the level of other environmental factors including air pollution [32,33], and therefore PM_10_ may play a role of mediator in the association between greenness and mortality. Further studies are required to examine the role of air pollution as a mediator in their association.

In the sensitivity analyses, we used various indicators to reflect complex dimensions of socioeconomic status, such as spending power, housing conditions, access to medical care, knowledge acquisition, and social resources [21,22]. Although these indicators are interrelated and using different indicators generally did not alter the association of PM_10_ and greenness with mortality, it is noteworthy to compare results to assess the validity of using single indicators, as there is no gold standard for representing socioeconomic status. In the city-specific analyses, some cities were found to have different patterns of association of PM_10_ or greenness with cause-specific mortality. Further studies are needed to explore spatial variations across cities and factors that contribute to these variations.

We observed lower risks of PM_10_ on cardiovascular mortality in greener areas. Systemic oxidative stress and inflammation have been suggested as the underlying mechanisms in which particulate matter causes adverse health effects [4,34]. Additionally, green space is known to encourage physical activity [8,9], which leads to increase antioxidant capacity [35,36] and induces anti-inflammatory response [37,38]. Greenness is also suggested to ameliorate stress through experience in green space or green views [11,39], which may prevent oxidative stress-related diseases including cardiovascular diseases. Therefore, PM_10_ and greenness have adverse and beneficial effects on the cardiovascular system, respectively, and the responses of exposure to air pollution and greenness may interact.

However, contrary to our expectations, we observed higher risks of PM_10_ on non-accidental mortality in greener areas. Kioumourtzoglou et al. [13] and de Keijzer et al. [14] reported stronger effects of PM_10_ on all-cause mortality with higher green space level. MacNaughton et al. [15] also found a larger effect of particulate matter in greener areas, although this study focused on school absenteeism as a health outcome. One possible explanation would be that greenness might have potential side effects on health. Previous studies suggested that greenness may produce allergenic pollens [40,41] and these pollens interact with air pollutants, thereby triggering or exacerbating respiratory disease [41,42,43]. Therefore, people living in greener areas could be more susceptible to air pollution, especially in terms of respiratory-related disease, and similar trends observed in chronic lower respiratory disease mortality seem to partially support this explanation. Similarly, green space may have other potential side effects on health by increasing exposure to pesticides, transmitting infections by arthropod vectors, or resulting in excessive exposure to UV radiation [44,45], which could rather reduce the immune defenses of people living in greener areas. However, studies regarding the adverse health impacts of greenness are inconclusive, and therefore more in-depth studies are needed to understand the underlying mechanisms of combined exposures to air pollution and greenness.

Our study has several limitations. First, given the inherent limitation of ecological study design, we could not obtain individual exposure data. Instead, we used the single representative exposure value in each district, assuming the exposure levels to PM_10_ and greenness are uniform for all residents living in the same district. However, if this assumption is violated due to short- (i.e., traveling outside the residential area for work or other purposes) or long-term (i.e., migration) population movements, it may result in exposure misclassification. Although we could not access any individual information for short- and long-term population movements, this exposure misclassification would be expected to be non-differential, which would bias our results towards the null [46]. Secondly, there is the possibility of residential self-selection, in that people who are more educated, health-conscious, or generally healthier might choose to live in eco-friendly neighborhoods [47]. However, in Korea, education settings and transportation infrastructure are the primary determinants in the choice of residential location [48]. Therefore, we expect that such self-selection would occur randomly among districts and potential bias due to residential self-selection did not affect our conclusion. Thirdly, NDVI does not distinguish the type, accessibility, and quality of greenness, which might be an important determinant of the type of contact with green space [49,50,51]. Future studies are needed to specify the different characteristics of green space to capture the effects of different types of greenness on health. Lastly, we could not include PM_2.5_ in the analyses as PM_2.5_ concentrations have been measured on a regular basis in nationwide since 2015.

Although we acknowledge these limitations, this study has several important implications. To our knowledge, it is the first study to examine the long-term effects of greenness on mortality in Korea. In addition, this study adds to the additional evidence for the potential role of green space in modifying the effects of air pollution. Our findings of the heterogeneous effects of greenness on the association between PM_10_ and cause-specific mortality suggest further research on the role of greenness across study areas with diverse characteristics of green space, air quality, and populations. As the urban population of the world has grown rapidly, it is important to understand health risks within a complex urban environment. In this respect, this study provides the scientific basis for understanding the potential risks and benefits of air pollution and greenness and the effects of their combined exposure.

## 5. Conclusions

The results of this study suggest that greenness may modify the association between PM_10_ and mortality, but this association differentially depends on the cause of death. Given the potential beneficial and detrimental effects of these environmental factors, further studies are needed to understand how the complex urban environment affects public health and its underlying mechanisms.

## Figures and Tables

**Table 1 ijerph-16-01866-t001:** Descriptive statistics for study variables in seven metropolitan cities, 2008–2016.

Variable	Mean	Median	Standard Deviation	25th–75th Percentile
PM_10_ (μg/m^3^)	47.65	47.13	6.48	43.00–51.88
NDVI	0.48	0.48	0.13	0.38–0.58
Percentage of adults with low education (%)	14.13	13.59	4.76	11.09–16.39
Smoking rate (%)	23.97	24.00	2.90	22.1–25.9
Health care resource index	0.00	−0.75	2.70	−1.60–0.54
Standardized mortality rates (per 100,000)				
Non-accidental	336.42	334.05	50.65	299.73–371.13
Cardiovascular disease	85.08	83.93	22.27	67.05–101.06
Ischemic heart disease	21.08	19.86	7.16	15.96–24.98
Respiratory disease	28.14	28.16	6.59	23.40–32.15
Chronic lower respiratory disease	8.90	8.41	3.78	6.00–11.13
Lung cancer	22.90	22.65	4.29	20.11–25.54

**Table 2 ijerph-16-01866-t002:** Percent changes in cause-specific mortality and 95% confidence interval for 10 μg/m^3^ increase in PM_10_ and interquartile range (IQR) increase in NDVI.

	Single-Exposure Model ^a,b^	Two-Exposure Model ^a,b^
	Percent Increase (95% Confidence Interval)	Percent Increase (95% Confidence Interval)
Non-accidental		
PM_10_ (per 10 μg/m^3^)	4.50% (3.42%, 5.58%)	4.49% (3.41%, 5.57%)
NDVI (per IQR) ^c^	−0.59% (−1.85%, 0.69%)	−0.40% (−1.59%, 0.81%)
Cardiovascular disease		
PM_10_ (per 10 μg/m^3^)	9.75% (7.67%, 11.86%)	9.70% (7.64%, 11.81%)
NDVI (per IQR) ^c^	−2.89% (−5.18%, −0.53%)	−2.56% (−4.68%, −0.39%)
Ischemic heart disease		
PM_10_ (per 10 μg/m^3^)	7.5%9 (4.28%, 11.00%)	7.50% (4.19%, 10.90%)
NDVI (per IQR) ^c^	−3.64% (−7.08%, −0.06%)	−3.45% (−6.84%, 0.07%)
Respiratory disease		
PM_10_ (per 10 μg/m^3^)	−3.23% (−5.46%, −0.96%)	−3.12% (−5.36%, −0.83%)
NDVI (per IQR) ^c^	1.85% (−0.76%, 4.52%)	1.53% (−1.07%, 4.19%)
Chronic lower respiratory disease		
PM_10_ (per 10 μg/m^3^)	16.13% (11.52%, 20.92%)	16.03% (11.42%, 20.85%)
NDVI (per IQR) ^c^	−3.75% (−8.50%, 1.24%)	−3.41% (−7.97%, 1.39%)
Lung cancer		
PM_10_ (per 10 μg/m^3^)	2.93% (0.87%, 5.03%)	2.98% (0.92%, 5.08%)
NDVI (per IQR) ^c^	1.10% (−1.22%, 3.47%)	1.25% (−1.06%, 3.62%)

^a^ Adjusted for neighborhood socioeconomic status (percentage of adults with low education), smoking rate, and healthcare infrastructure status. ^b^ Models with correction for spatial autocorrelation. ^c^ IQR for NDVI = 0.20.

**Table 3 ijerph-16-01866-t003:** Percent changes in cause-specific mortality per 10 μg/m^3^ increase in PM_10_ by the level of greenness.

	Non-Accidental	Cardio Vascular Disease	Ischemic Heart Disease	Respiratory Disease	Chronic Lower Respiratory Disease	Lung Cancer
Greenness ^a^						
High	5.83% (3.95%, 7.74%)	7.46% (3.97%, 11.07%)	1.89% (−4.51%, 8.72%)	−1.27% (−5.12%, 2.73%)	20.88% (12.54%, 29.82%)	4.32% (0.19%, 8.62%)
Medium	3.57% (1.81%, 5.37%)	8.56% (5.05%, 12.18%)	6.73% (1.58%, 12.13%)	−3.14% (−6.78%, 0.64%)	14.09% (6.73%, 21.95%)	1.51% (−1.56%, 4.67%)
Low	3.45% (1.42%, 5.51%)	11.23% (7.28%, 15.32%)	7.86% (1.52%, 14.60%)	−9.23% (−12.83%, −5.47%)	11.20% (1.57%, 21.74%)	4.34% (0.26%, 8.59%)
*p*-value for interaction ^b^	0.01	0.67	0.10	0.18	0.47	0.45

^a^ Greenness was based on the NDVI value at each district level. A high group was defined as those with values of ≥66th percentile, a medium group was defined as those with values of ≥33th percentile, and a low group was defined as those with values of <33th percentile. ^b^
*p*-value for interaction (PM_10_ × greenness categories in tertiles).

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
