# Peer review of "Interactions between Ambient Air Particles and Greenness on Cause-specific Mortality in Seven Korean Metropolitan Cities, 2008–2016"

_ijerph, 2019, doi:10.3390/ijerph16101866_

Round 1
Reviewer 1 Report
General comment
This is an interesting population study investigating the interaction between PM10 exposures and greenness exposure on cause-specific disease mortality. Overall, the research methods are appropriate for addressing the research question. However, my main concern is that authors only highlighted the interaction results between PM10 and greenness on cardiovascular disease mortality, rather than the other disease mortality. The interaction results related to other disease mortality seem to have opposite direction than the cardiovascular disease. These inconsistent findings across different disease outcomes need to be fully discussed and cannot be ignored in the abstract. Also, results in other air pollution exposure such as NO2 and SO2 are not presented in this manuscript, which raises another question that why this study only focuses on PM10 exposure? More specific comments are described below.
Specific comments:
1. A better description of health care resource index is needed for the method. This index variable has largely 0 values and the standard deviation is so wide. Is this a good variable to be included in the model and whether the adjustment for this variable has a significant influence on the results?
2. As the authors mentioned, there could be a cohort effect across generations born in different years. Although the covariate variable for education was classified differently by two age groups (greater or lower than 45 years old), the cohort effect should be more carefully adjust for in the analysis.
3. On page 3, line 41, authors said: “Table 1 shows the descriptive statistics of cause-specific standardized mortality rates”. Please confirm whether the statistics presented in Table 1 are standardized mortality rates or statistics for variable distributions?
4. The presentation of effect sizes and 95% CIs are confusing. For example, on page 4, line 4, “…a 10 μg/m3 increase in annual average PM10 was associated with 4.49 % (95% Confidence Interval (CI): 3.41, 5.57)…”, whether the number included in the 95% CI here means 3.41% - 5.57%?
5. On page 5, line 11-12, the sentence was omitted a “for” in the end “The associations between PM10 and cause-specific mortality were robust when either SO2 or NO2 was additionally adjusted FOR”.
6. For table 3, it will be better to include interaction p-values in the table to show whether the interaction test is significant or not.
7. I feel the results presented in the abstract is somehow “cherry-picking”. There is a clearly opposite interaction between PM10 and greenness exposure in terms of the relationship with respiratory disease or non-accidental mortality compared to cardiovascular disease mortality. This opposite interaction effects warrants more discussion and be included in the abstract.
8. It will be good to have several figures describing the trend of diseases mortality and PM10 and greenness exposure over the study period 2008-2016. One question is whether the population trend of disease mortality and exposure concentration is supported by the findings of this study?
9. Since authors have the data for gaseous pollutants, why authors did not present association results for air pollutant such as NO2 and SO2, but only focused on PM10?
10. Have authors compared the association difference between urban and rural areas? If this study includes some population from rural areas, it will be interesting to explore this question and include the results in this manuscript.
Author Response
Dear Editor and Reviewers,
We appreciate the opportunity to revise our paper titled “Interactions between ambient air particles and greenness on cause-specific mortality in seven Korean metropolitan cities, 2008-2016”. We would like to thank the editor and reviewers for their critiques and constructive recommendations. We did our best to address all comments addressed by the reviewers. Generally, 1) we have revised the abstract and discussion section to highlight and discuss about interactions between PM10and greenness shown in the opposite direction depending on the cause of death, 2) we have added the additional descriptions to clarify the meaning of statements, 3) we have added the additional results suggested by reviewers in both manuscript and supplementary materials, and 4) we have corrected typos and grammatical errors. The specific revisions made in response to the reviewer are detailed below.
# Reviewer 1
Comment 1
A better description of health care resource index is needed for the method. This index variable has largely 0 values and the standard deviation is so wide. Is this a good variable to be included in the model and whether the adjustment for this variable has a significant influence on the results?
Response to Comment 1
The health care resource index is the sum of the z-scores of three indicators (number of medical personnel, hospitals, and hospital beds), which consequently has a mean of 0 (note that this does not mean that there are largely 0 values). The standard deviation was 2.70, which is very natural because these three indicators are not expected to be independent. Generally, the use of variable whose variance is large is expected to have high statistical power to detect the association between entity we wish to measure (by using the variable) and health outcome.
As South Korea has experienced remarkable advances in medical care quality and accessibility that contributed to improvement in health over decades [1], we included this index in the model to account for a secular trend in improving health care resource. The figure below shows that three indicators from this index reflect the increasing trend of health care resource well, and the index had a significant influence on the results as expected.
We have added a description on health care resource index and clarified our process for calculating the index in the methods section. The details of the revision are displayed below.
Figure.Trends of the number of medical personnel, hospitals, and hospital beds in seven Korean metropolitan cities
Page 3, lines 6-13
The health care resource index was used to indicate neighborhood healthcare infrastructure status, using data from the 2008-2016 National Health Insurance Service. This index consisted of the numbers of medical personnel, hospitals, and hospital beds (per 100,000 population), which are widely used indicators to represent and compare health care resources across countries [24]. We converted the value on each indicator to a z-score and used the sum of the z-scores as the health care resource index. Therefore, the index has a mean of 0 and a higher value of the index indicates a higher utilization of health care resources.
Comment 2
As the authors mentioned, there could be a cohort effect across generations born in different years. Although the covariate variable for education was classified differently by two age groups (greater or lower than 45 years old), the cohort effect should be more carefully adjust for in the analysis.
Response to Comment 2
If we understand correctly, the reviewer points out the need for adjusting the cohort effects in investigating the association between exposure and health outcome.
The sentence “to consider the different meaning of education level for different birth cohorts (Page 3, lines 95-96 in the original manuscript)” was meant to consider the different level of educational attainment between birth cohorts in measuring the education variable, not the potential cohort effects in investigating the associations of PM10and greenness with mortality (note that we think there is a low possibility of cohort effect in our study population and the purpose of this study was to investigate their association in general). In Korea, there have been considerable changes in education opportunity[2]. Therefore, if we apply the same standard for low education (e.g., less than high school education), older population will be overrepresented among those classified as less educated even if some of them are actually at higher SES (i.e. measurement error). We also conducted sensitivity analyses using different SES indicators to evaluate the validity of education indicator we used, and the results were consistent.
We have revised descriptions to clarify it. The details of the revision are displayed below.
Page 2, line 49 – Page 3, line 5
We measured the percentage of adults with low education using the Population and Housing Census. Low education was defined as 1) less than high school education for those aged under 45, and 2) less than middle school education for those aged 45 or over [23]. By using a different standard of low education according to age groups, we intend to avoid the problem that older population, whose educational attainment are relatively lower than younger population, may be overrepresented as less educated.
Comment 3
On page 3, line 41, authors said: “Table 1 shows the descriptive statistics of cause-specific standardized mortality rates.” Please confirm whether the statistics presented in Table 1 are standardized mortality rates or statistics for variable distributions?
Response to Comment 3
We have modified the description to clarify the contents of Table 1. The details of the revision are displayed below.
Page 3, lines 42-43
Table 1 presents the descriptive statistics of the study variables.
Page 3, line 41 (in original script)
Table 1 shows the descriptive statistics of cause-specific standardized mortality rates. (deleted)
Comment 4
The presentation of effect sizes and 95% CIs are confusing. For example, on page 4, line 4, “… a 10mg/m3increase in annual PM10 was associated with 4.49 % (95% Confidence Interval (CI): 3.41, 5.57) …”, whether the number included in the 95% CI here means 3.41% - 5.57%?
Response to Comment 4
We have modified the way to present 95% confidence intervals (throughout the manuscript). The example of the revision is displayed below.
Page 4, lines 6-7
… a 10mg/m3increase in annual average PM10was associated with 4.49 % (95% Confidence Interval (CI): 3.41%, 5.57%) increase in non-accidental …
Comment 5
On page 5, line 11-12, the sentence was omitted a “for” in the end “The associations between PM10 and cause-specific mortality were robust when either SO2 or NO2 was additionally adjust FOR”.
Response to Comment 5
We have corrected this grammatical error. The details of the revision are displayed below.
Page 5, lines 12-13
The associations between PM10and cause-specific mortality were robust when either SO2or NO2was additionally adjusted for [Table S2].
Comment 6
For table 3, it will be better to include interaction p-values in the table to show whether the interaction test is significant or not.
Response to Comment 6
Thank you for the valuable suggestion. We have added interaction p-values to Table 3 to show whether the interaction term is significant or not (Page 6, Table 3).
Comment 7
I feel the results presented in the abstract is somehow “cherry-picking”. There is a clearly opposite interaction between PM10 and greenness exposure in terms of the relationship with respiratory disease or non-accidental mortality compared to cardiovascular disease mortality. This opposite interaction effects warrants more discussion and be included in the abstract.
Response to Comment 7
We have revised the abstract to incorporate the reviewer’s recommendation. The details of the revision are displayed below. And we believe the statements provided in page 7, lines 19-34 give the relevant discussion about the potential explanation of observed interaction in non-accidental and chronic lower respiratory disease mortality. Also, the statements provided in page 8, lines 4-6 include a suggestion to explore the role of greenness across diverse study areas to understand the heterogeneous interactions depending on the cause of death.
Page 1(Abstract), lines 23-27
In the stratified analyses, we found interactions between PM10and greenness, but in the opposite direction depending on the cause of death. The effects of PM10on cardiovascular-related mortality were attenuated in greener areas, whereas the effects of PM10on non-accidental mortality were attenuated in less green area. Further studies are needed to explore the underlying mechanisms.
Comment 8
It will be good to have several figures describing the trend of diseases mortality and PM10 and greenness exposure over the study period 2008-2016. One question is whether the population trend of disease mortality and exposure concentration is supported by the findings of this study?
Response to Comment 8
Thank you for the valuable suggestion. We have added figures describing the annual trend of cause-specific mortality as well as main exposure variable (PM10and greenness indicated by NDVI) over the study period in the Supplementary materials (Figure S10-17). Both cause-specific standardized mortality rates, except respiratory disease mortality, and the concentration of PM10are decreasing over time, which is supported by the findings of our study. An increasing trend in respiratory disease mortality seems to be affected by the increasing trend of pneumonia mortality. This trend implicates that there would be other risk factors of respiratory mortality (other than PM10), in other words, residual confounding by unknown factor, and we mentioned about this in the discussion section (Page 6, lines 19~).
Comment 9
Since authors have the data for gaseous pollutants, why authors did not present association results for air pollutant such as NO2 and SO2, but only focused on PM10?
Response to Comment 9
As the title “Interactions between ambient air particles and greenness on cause-specific mortality in seven Korean metropolitan cities, 2008-2016” implies, the main interest of this study was PM10, which was motivated by the fact that PM has been known to be the strongest and most consistent associations with various health outcomes among the criteria air pollutants (PM2.5data was not available for the study period).
As the reviewer pointed it out, the results for NO2and SO2will give additional insight into the health effects of gaseous pollutants and their interactions with greenness.Therefore, the results for the associations between gaseous pollutants and cause-specific mortality have been added in the Supplementary materials (Table S5). We did not conduct further analyses for evaluating interactions between NO2/SO2and greenness because another in-depth research is needed to account for different characteristics of gaseous pollutants (compared to PM). Future studies would need to explore their interactions.
Comment 10
Have authors compared the association difference between urban and rural areas? If this study includes some population from rural areas, it will be interesting to explore this question and include the results in this manuscript.
Response to Comment 10
Thank you for the valuable suggestion. We agree that it will be a valuable study to compare the association between urban and rural areas because the characteristics of the study population and environmental exposures (PM and greenness) in urban/rural are different. However, as we described the characteristics of the study area (Page 3, lines 40-41), our study areas are mainly classified as urban areas where (on average) 97.72% people in each district live in urban area. Thus, with the purpose of understanding health risks within a complex urban environment (Page 8, lines 7-8), we focused on urban areas in this study.
To explore the difference between urban and rural areas, a more expanded study area (such as nationwide) will be needed. Therefore, further research will be required to explore this question.
Reference
1. Yang, S., Khang, Y. H., Harper, S., Davey Smith, G., Leon, D. A., & Lynch, J. (2010). Understanding the rapid increase in life expectancy in South Korea. American journal of public health, 100(5), 896-903.
2. Choi, Y. J., Jeong, B. G., Cho, S. I., Jung-Choi, K., Jang, S. N., Kang, M., & Khang, Y. H. (2007). A Review on Socioeconomic Position Indicators in Health Inequality Research. Journal of Preventive Medicine and Public Health, 40(6), 475-486. (in Korean)

Reviewer 2 Report
This is an interesting study to examine the combined effects of air pollution and greenness on cause-specific mortality. I provided a few comments for the authors to consider:
1. The definition of PM10 was wrong, smaller than 10 um, not 10 ug/m3
2. For the association, please specify what unit for?
3. Are there any reporting differences among difference areas?
4. Any other confounding factors were controlled for? For example, socioeconomic, health care?
5. Any statistical testing was conducted to test the statistical significance of the stratified analyses?
Author Response
Dear Editor and Reviewers,
We appreciate the opportunity to revise our paper titled “Interactions between ambient air particles and greenness on cause-specific mortality in seven Korean metropolitan cities, 2008-2016”. We would like to thank the editor and reviewers for their critiques and constructive recommendations. We did our best to address all comments addressed by the reviewers. Generally, 1) we have revised the abstract and discussion section to highlight and discuss about interactions between PM10and greenness shown in the opposite direction depending on the cause of death, 2) we have added the additional descriptions to clarify the meaning of statements, 3) we have added the additional results suggested by reviewers in both manuscript and supplementary materials, and 4) we have corrected typos and grammatical errors. The specific revisions made in response to the reviewer are detailed below.
# Reviewer 2
Comment 1
The definition of PM10 was wrong, smaller than 10mg, not 10mg/m3
Response to Comment 1
Thank you for the comment. We have corrected the definition of PM10. The details of the revision are displayed below.
Page 1(Abstract), lines 13-14
This study aims to investigate the association of particulate matter with an aerodynamic diameter smaller than 10 mm (PM10) and (…)
Page 2, lines 13-14
This study investigates the effects of particulate matter with an aerodynamic diameter smaller than 10 μm (PM10) and (…)
Comment 2
For the association, please specify what unit for?
Response to Comment 2
We believe the statements provided in Page 3, lines 28-30 in the original manuscript specified the unit for the association of PM10and greenness with mortality, respectively. In the results section, their units were also mentioned.
Page 3, lines 28-30 (Page 3, lines 33-34 in revised manuscript)
The results for the association of PM10and greenness with mortality were presented for 10 μg/m3and interquartile range (IQR) increase, respectively.
Comment 3
Are there any reporting differences among difference areas?
Response to Comment 3
We have added results for city-specific associations of PM10and greenness with mortality to report differences among different cities in Supplemental materials (Figure S18-19). We have also added a relevant discussion of these results in the discussion section.
Page 3, lines 32-34 (methods)
We also conducted city-specific analyses to explore spatial variations in the association with PM10and greenness with mortality.
Page 5, lines 15-20 (results)
In the city-specific analyses, we found different patterns of the association of PM10or greenness with mortality across cities [Figure S18-19]. Daejeon was found to have a negative estimate in the association with PM10and cardiovascular and respiratory-related mortality. And Daegu and Ulsan were found to have positive, but non-significant estimates in the association with greenness and ischaemic heart disease mortality.
Page 7, lines 6-9 (discussion)
In the city-specific analyses, some cities were found to have different patterns of the association of PM10or greenness with cause-specific mortality. Further studies are needed to explore spatial variations across cities and factors that contribute to these variations.
Comment 4
Any other confounding factors were controlled for? For example, socioeconomic, health care?
Response to Comment 4
We controlled for confounders including district-level socioeconomic status (indicated by the percentage of adults with low education in main analyses), smoking rate, healthcare infrastructure status (indicated by health care resource index), and gaseous pollutants (NO2or SO2) in the analyses. We believe the statements provided in Page 2, line 45 – Page 3, line 12 include a detailed description for confounding factors.
We also considered potential confounders related to health behaviors, such as the proportion of those who are physically active or drinking alcohol. However, we did not include them in the final model because those variables did not have a significant influence on the results.
Comment 5
Any statistical testing was conducted to test the statistical significance of the stratified analyses?
Response to Comment 5
Thank you for the valuable suggestion. We have added interaction p-values to Table 3 to show whether the interaction term is significant or not (Page 6, Table 3). We have also added a description in the methods section.
Page 3, lines 24-26
To investigate whether the association between PM10and mortality varied by the level of greenness, we included an interaction term between PM10and greenness in the model and stratified the analyses.

Reviewer 3 Report
This ecological study, at 74 districts in 7 Korean metropolitan cities, uses data from 1-2 monitoring station per district to calculate annual average PM10 for the time series. Satellite-derived NDVI greenness averaged for the summer season is also calculated. Registry data were used to determine counts for several causes of deaths (non-accidental, CVD, IHD, respiratory disease and lung cancer as negative control). Sensitivity analyses including gaseous pollutants and other single indicators for SES were included. Interestingly, the analysis is also stratified by districts with low, medium or high levels of greenness and the results show that greenness may modify the associations (differentially) for the investigated outcomes. The paper is well written and presented. I have only a few comments or questions that need clarification.
1. The reasons for selecting education as the main SES variable are given. But do the two citations both support the notion that education is the best single indicator, and are they applicable to the Korean situation? The authors constructed an index for health care resources. It might be interesting to develop and use an SES index including some of the other (not highly correlated) available SES indicators.
2. If correction for spatial autocorrelation was needed, what district coordinate was used? A geographic centroid, population weighted centroid, something else? Also, it is not mentioned in the results whether any of the models need to be treated for spatial autocorrelation.
3. I understand the PM10 data are annual means, by district, for the study period. What about NDVI, is this also available as the annual means or just as a single long-term average? (pg 3, line 38).
4. What does this mean on page 5, line 2: “but their interactions were shown in the opposite direction by the cause of death”?
5. The authors note the attenuation for NDVI in the 2 pollutant models in the results. But unless I missed it, I don’t think the authors comment on this in the discussion yet. Please add.
6. I would not read too much into the sensitivity analysis on respiratory disease which is currently mentioned twice (pg 6, line 18 & 43). Only the 3rd tested alternative SES variable is non-significant. Also for NO2 the upper confidence interval is only just over 0.
7. Table 2 footnote should mention the exact SES indicator as neighbourhood SES is too vague (in this study where several are explored).
8. The first sentence of the conclusion needs to be carefully written. It essentially says that low levels of greenness are associated with risk of mortality. This is not true; I think the authors rather intend to point out the interaction. The abstract line 22 also implies this and should be fixed.
9. It is nice to see the maps in the supplement, but as presented they are not helpful. Either the map of Korea should be the base, showing the 7 cities (5 areas) where they actually belong. Or if the intention is to to facilitate comparison of the different exposures and outcomes by area, a grid style presentation of 5 pages (one for each area) with 8 small maps (eg. showing the info from Fig S1-S8 by study area) would work.
Author Response
Dear Editor and Reviewers,
We appreciate the opportunity to revise our paper titled “Interactions between ambient air particles and greenness on cause-specific mortality in seven Korean metropolitan cities, 2008-2016”. We would like to thank the editor and reviewers for their critiques and constructive recommendations. We did our best to address all comments addressed by the reviewers. Generally, 1) we have revised the abstract and discussion section to highlight and discuss about interactions between PM10and greenness shown in the opposite direction depending on the cause of death, 2) we have added the additional descriptions to clarify the meaning of statements, 3) we have added the additional results suggested by reviewers in both manuscript and supplementary materials, and 4) we have corrected typos and grammatical errors. The specific revisions made in response to the reviewer are detailed below.
# Reviewer 3
Comment 1
The purpose for selecting education as the main SES variable are given. But do the two citations both support the notion that education is best single indicator, and are they applicable to the Korean situation? The authors constructed an index for health care resources. It might be interesting to develop and use an SES index including some of the other (not highly correlated) available SES indicators.
Response to Comment 1
Education and occupation are indicators which are most widely used in Korean research [1, 2]. There have been many Korean studies showing that education has independent effects on health outcomes (even after adjusting for other SES indicators) and some studies reported the dominance effect of education over that of occupation [3]. Also, educational attainment in Korea is highly correlated with occupation, income levels, and social mobility. The problem that can arise when we apply education indicator in Korean situation is the different meaning of education level for different birth cohorts [1, 4] and we considered it by using different standards according to age groups. In addition, we conducted sensitivity analyses to compare the results using different SES variables. And we believe that similar results across different SES indicators lend confidence to the validity of education indicator.
We appreciate the reviewer’s recommendation to develop a SES index, which will be a reasonable next research step and we will include it in a future study.
Comment 2
If correction for spatial autocorrelation was needed, what district coordinate was used? A geographic centroid, population weighted centroid, something else? Also, it is not mentioned in the results whether any of the models need to be treated for spatial autocorrelation.
Response to Comment 2
We have added an additional description to describe which district coordinate was used and which model was adjusted for spatial autocorrelation in the methods section. We have also clarified this on Table 2. The details of the revision are displayed below.
Page 3, lines 20-22
Spatial autocorrelation of residuals was tested using Moran’s I statistics. As we found significant residual spatial autocorrelation in all models, we additionally adjusted for the coordinate of each district in the model. We used the coordinates of each district’s office station.
Comment 3
I understand the PM10 data are annual means, by district, for the study period. What about NDVI, is this also available as the annual means or just as a single long-term average? (pg3, line 38).
Response to Comment 3
We used the annual NDVI value (median value for each year; 2008-2016). We believe the statement provided in Page 2, line 41 include the information about the time period of data we obtained. And we have revised the statement in the abstract to clarify it. We have also added the annual trend of NDVI in the Supplementary materials [Figure S11] so that the readers can see the annual value of NDVI.
Page 1(Abstract), line 16
We obtained the annual standardized cause-specific mortality rates, annual mean concentration of PM10, and annual Normalized Difference Vegetation Index (NDVI) for 73 districts for the period 2008-2016.
Comment 4
What does this mean on page 5, line 2: “but their interactions were shown in the opposite direction by the cause of death”?
Response to Comment 4
The sentence “but their interactions were shown in the opposite direction by the cause of death” was meant to highlight the opposite interaction observed in non-accidental mortality and cardiovascular-related mortality. We have revised the sentence to clarify it. The details of the revision are displayed below.
Page 5, lines 2-3
The associations between PM10and mortality varied by the level of greenness. However, greenness modified the associations differentially depending on the cause of death.
Comment 5
The authors note the attenuation for NDVI in the 2 pollutant models in the results. But unless I missed it, I don’t think the authors comment on this in the discussion yet. Please add.
Response to Comment 5
We have added an additional explanation on results in addition to statements in the original manuscript. The details of the revision are displayed below.
Page 6, lines 36-42
We also additionally adjusted for PM10to control for its indirect effects on health impacts. After adjusting for PM10, the associations between greenness and mortality were slightly attenuated, which indicates that PM10was responsible for some of the effects of greenness on mortality. Greenness is suggested to reduce the level of other environmental factors including air pollution [32, 33], and therefore PM10may play a role of mediator in the association between greenness and mortality. Further studies are required to examine the role of air pollution as a mediator in their association.
Comment 6
I would not read too much into the sensitivity analysis on respiratory disease which is currently mentioned twice (pg6, line 18 & 43). Only the 3rdtested alternative SES variable is non-significant. Also for NO2 the upper confidence interval is only just over 0.
Response to Comment 6
We have taken the reviewer’s suggestion and deleted those two statements.
Comment 7
Table 2 footnote should mention the exact SES indicator as neighbourhood SES is too vague (in this study where several are explored).
Response to Comment 7
We have clarified which SES indicator was used in the main analyses in Table 2. The details of the revision are displayed below.
Page 5, Table 2 (footnote)
a Adjusted for neighborhood socioeconomic status(percentage of adults with low education), smoking rate, and healthcare infrastructure status.
Comment 8
The first sentence of the conclusion needs to be carefully written. It essentially says that low levels of greenness are associated with risk of mortality. This is not true; I think the authors rather intend to point out the interaction. The abstract line 22 also implies this and should be fixed.
Response to Comment 8
Thank you for the valuable suggestion. We have taken the reviewer’s recommendation and deleted the first sentence. Instead, we have revised the conclusion to focus on interactions between PM10and greenness. We have also revised the abstract to clarify our results on the association between greenness and cause-specific mortality. The details of the revision are displayed below.
Page 1(Abstract), lines 21-23
Our findings suggest an increased risk of all causes examined, except respiratory disease mortality, with high levels of PM10and decreased risk of cardiovascular-related mortality with a high level of greenness.
Page 8, lines 12-15
The results of this study suggest that greenness may modify the association between PM10and mortality, but differentially depending on the cause of death. Given the potential beneficial and detrimental effects of these environmental factors, further studies are needed to understand how the complex urban environment affects public health and its underlying mechanisms.
Comment 9
It is nice to see the maps in the supplement, but as presented they are not helpful. Either the map of Korea should be the base, showing the 7 cities (5 areas) where they actually belong. Or if the intention is to facilitate comparison of the different exposures and outcomes by area, a grid style presentation of 5 pages (one for each area) with 8 small maps (eg. showing the info from Fig S1-S8 by study area) would work.
Response to Comment 9
Thank you for the valuable suggestion. We have added the map of Korea showing the location of seven cities in the supplementary materials (Figure S1).
Reference
1. Choi, Y. J., Jeong, B. G., Cho, S. I., Jung-Choi, K., Jang, S. N., Kang, M., & Khang, Y. H. (2007). A Review on Socioeconomic Position Indicators in Health Inequality Research. Journal of Preventive Medicine and Public Health, 40(6), 475-486. (in Korean)
2. Khang, Y. H., & Kim, H. R. (2005). Relationship of education, occupation, and income with mortality in a representative longitudinal study of South Korea. European journal of epidemiology, 20(3), 217-220.
3. Son, M., Armstrong, B., Choi, J. M., & Yoon, T. Y. (2002). Relation of occupational class and education with mortality in Korea. Journal of Epidemiology & Community Health, 56(10), 798-799.
4. Kim, Y. M.; Kim, M. H., Health inequalities in Korea: current conditions and implications. Journal of preventive medicine and public health = Yebang Uihakhoe chi 2007, 40 (6), 431-8.

Round 2
Reviewer 1 Report
The authors have addressed most of my comments. I have three additional comments: 1. The cohort effect has not been adjusted for in the analysis. Whether the air pollution exposure and greenness exposure can be varied by different generations, which may influence the final association results between the exposure and mortality rates? Variables such as age or birth year should be included in the covariate adjustment. 2. There are no significant interactions between greenness and PM10 exposure in terms of cardio-vascular disease or ischemic heart disease mortality, as the Table 3 shows. However, this was presented as significant and main result in the abstract and the result section. Authors seem to overstated these results. 3. I did not see supplementary tables in your previous email. I would like to review Table S5 to see whether the results of NO2/NOx are worth to be included in this manuscript. I am not convinced yet why the study was focused on PM10 exposure, rather than fine particles PM2.5 or gaseous pollutants, which may have more direct influence on the biological system.
Author Response
Response to Comment 1
Thank you for the valuable comment. We hardly think that air pollution exposure and greenness exposure are varied by different generations, because the occurrence of different exposure patterns between generations during the study exposure period, which is a 1-year, is hard to imagine. As indicated by the reviewer, if the cohort effects exist, it needs to be considered. And adjusting age or birth year in the model would be one of the solutions. This study is an ecological study so that we used the age-standardized (using 5-year age interval) mortality rates (Page 2, lines 26-28), which gives summary measures that are independent of age composition and implicitly adjusts for the potential cohort effects. If the reviewer has another suggestion on this issue, we are open to consideration of any further comment.
Comment 2
There are no significant interactions between greenness and PM10 exposure in terms of cardiovascular disease or ischemic heart disease mortality, as the Table 3 shows. However, this was presented as significant and main result in the abstract and the result section. Authors seem to overstated these results.
Response to Comment 2
As indicated by the reviewer, we addressed the interactions between PM10and greenness in cardiovascular-related mortality as the main results together with interaction in non-accidental mortality. Even though the p-value for interaction terms in cardiovascular-related mortality was higher than 0.05 (as often done), it is not an evidence of the absence of interaction [1 - 3]. And, practically, the statistical power to detect interaction is lower than for main effects [3, 4]. More to the point, we observed distinct patterns of the association between PM10and cardiovascular-related mortality across the level of greenness. We also discussed a biologically plausible explanation supporting their relationships in the manuscript.
However, we agree with the reviewer that the statistical significance of interaction terms on cardiovascular-related mortality should be mentioned in the manuscript to clarify the results. Thus, we have revised the manuscript. The details of the revision are displayed below.
Page 5, lines 11-12
However, the interaction terms in cardiovascular and ischaemic heart disease mortality were not statistically significant (p=0.67 and 0.10, respectively).
Page 6, lines 7-10
The effect of PM10on non-accidental mortality was stronger in districts with a higher level of greenness. On the other hand, the effect of PM10on cardiovascular mortality was stronger in districts with a lower level of greenness, although interaction terms were not significant.
Reference
1. Matthews, J. N., & Altman, D. G. (1996). Statistics Notes: Interaction 2: compare effect sizes not P values. BmJ, 313(7060), 808.
2. Greenland, S., Senn, S. J., Rothman, K. J., Carlin, J. B., Poole, C., Goodman, S. N., & Altman, D. G. (2016). Statistical tests, P values, confidence intervals, and power: a guide to misinterpretations. European journal of epidemiology, 31(4), 337-350.
3. Altman, D. G., & Bland, J. M. (2003). Interaction revisited: the difference between two estimates. Bmj, 326(7382), 219.
4. VanderWeele, T. J., & Knol, M. J. (2014). A tutorial on interaction. Epidemiologic Methods, 3(1), 33-72.
Comment 3
I did not see supplementary tables in your previous email. I would like to review Table S5 to see whether the results of NO2/NOx are worth to be included in this manuscript. I am not convinced yet why the study was focused on PM10 exposure, rather than fine particles PM2.5 or gaseous pollutants, which may have more direct influence on the biological system.
Response to Comment 3
We have added the Table S5 below (and the reviewer can also see this table in the supplementary materials).
Table S5. Percent changes in cause-specific mortality and 95 % confidence interval for IQR increase in NO2and SO2in single- and two-pollutant model
Non-accidental | Cardio vascular disease | Ischaemic heart disease | Respiratory disease | Chronic lower respiratory disease | Lung cancer | |
Single pollutant (NO2)a | 3.55 (1.82, 5.32) | 8.09 (4.76, 11.53) | 6.18 (1.15, 11.46) | -5.09 (-8.42, -1.63) | 12.56 (5.27, 20.36) | 1.56 (-1.54, 4.76) |
+ PM10 | 1.81 (0.13, 3.53) | 4.20 (1.09, 7.41) | 3.29 (-1.69, 8.53) | -4.00 (-7.52, -0.34) | 5.63 (-1.18, 12.91) | 0.41 (-2.75, 3.66) |
Single pollutant (SO2)b | 1.12 (0.01, 2.23) | 1.10 (-0.95, 3.20) | 0.57 (-2.52, 3.77) | -0.47 (-2.73, 1.84) | 6.10 (1.65, 10.75) | 2.16 (0.14, 4.22) |
+ PM10 | 0.98 (-0.04, 2.02) | 0.80 (-1.08, 2.72) | 0.47 (-2.63, 3.67) | -0.38 (-2.63, 1.93) | 5.01 (0.92, 9.26) | 2.25 (0.22, 4.32) |
a IQR for NO2 = 10.35 b IQR for SO2 = 2.10 |
First of all, the primary interest of this study was PM10, which was motivated by the fact that PM has been known to be the strongest and most consistent associations with various health outcomes among the criteria air pollutants [1, 2]. And, as indicated by the reviewer, PM2.5is considered as a stronger risk factor than PM10. However, PM2.5concentrations have been measured on a regular basis in nationwide since 2015 so that we could not include PM2.5in this study. Thus, we have added the sentence to clarify it. The details of the revision are displayed below.
For the gaseous pollutants, there are controversies for their long-term health effects in current levels. In addition, it is still unclear as to whether these gaseous pollutants themselves are responsible for the observed adverse effects, or they are surrogate for fine particles or other correlated pollutants [2 - 6]. It has been discussed that the NO2associations may reflect the effects of other highly correlated air pollutants, mainly PM or other components of traffic-related air pollutants [4, 5]. For the SO2, the EPA revoked the annual and 24-hour SO2standards due to the lack of evidence on its health effects [6].
In Table S5, we observed that the estimates for the associations between NO2/ SO2and cause-specific mortality were attenuated and most of the estimates became statistically insignificant with adjustment for PM10. Taken together, there is insufficient evidence to indicate the effects of gaseous pollutants and no strong rationale to further investigate the interaction between these gaseous pollutants and greenness. Thus, we did not include gaseous pollutants in the main analyses and focused on PM10in this study.
Page 8, lines 5-6
Lastly, we could not include PM2.5in the analyses as PM2.5concentrations have been measured on a regular basis in nationwide since 2015.
Reference
1. Pope III, C. A. (2000). Epidemiological basis for particulate air pollution health standards. Aerosol Science & Technology, 32(1), 4-14.
2. World Health Organization. (2006). Air quality guidelines: global update 2005: particulate matter, ozone, nitrogen dioxide, and sulfur dioxide. World Health Organization.
3. Sarnat, J. A., Schwartz, J., Catalano, P. J., & Suh, H. H. (2001). Gaseous pollutants in particulate matter epidemiology: confounders or surrogates?. Environmental health perspectives, 109(10), 1053-1061.
4. Mills, I. C., Atkinson, R. W., Anderson, H. R., Maynard, R. L., & Strachan, D. P. (2016). Distinguishing the associations between daily mortality and hospital admissions and nitrogen dioxide from those of particulate matter: a systematic review and meta-analysis. Bmj Open, 6(7), e010751.
5. U.S. EPA. Integrated Science Assessment (ISA) for Oxides of Nitrogen – Health Criteria (Second External Review Draft, 2015). U.S. Environmental Protection Agency, Washington, DC, EPA/600/R-14/006, 2015.
6. U.S. EPA. (2010). Primary National Ambient Air Quality Standard for Sulfur Dioxide; Final Rule. 40 CFR Parts 50, 53, and 58.

Reviewer 2 Report
I have a few minor revisions:
1. The definition of PM10 on Page 1 is still wrong.
2. For the effects of particulate pollution, a few important references should be discussed.
Author Response
# Reviewer 2
Comment 1
The definition of PM10 on Page 1 is still wrong.
Response to Comment 1
As indicated by the reviewer, we corrected the definition of PM10(in Page 1 and Page 2) in first revised manuscript. We rechecked the manuscript and could not find any other error. If we missed any error, please let us know. (And we found that the micrometer could be shown wrongly in the review report on website. Please check the attached file.)
Page 1(Abstract), lines 13-14
This study aims to investigate the association of particulate matter with an aerodynamic diameter smaller than 10 (PM10) and (…)
Page 2, lines 13-14
This study investigates the effects of particulate matter with an aerodynamic diameter smaller than 10 (PM10) and (…)
Comment 2
For the effects of particulate pollution, a few important references should be discussed.
Response to Comment 2
Following the reviewer’s suggestion, we have added the biological mechanism of particulate matter on health and the result of meta-analysis for long-term effects of PM10in discussion section. The details of the revision are displayed below.
Page 6, lines 12-17
While the underlying biological mechanisms are unclear, exposure to ambient particulate matter has been related to the occurrence of pulmonary and systemic oxidative stress and inflammation, perturbation of autonomic nervous system balance, and translocation of particulate matter directly into the systemic circulation [1, 4]. A meta-analysis by Hoek et al. [3] reported the excess risk of 3.5 % (95 % CI: 0.4%, 6.6%) per 10 μg/m3increase in PM10for all-cause mortality.
